# Lectin-like Transcript-1 (LLT1) Expression in Oral Squamous Cell Carcinomas: Prognostic Significance and Relationship with the Tumor Immune Microenvironment

**DOI:** 10.3390/ijms25084314

**Published:** 2024-04-13

**Authors:** Juan C. de Vicente, Paloma Lequerica-Fernández, Juan P. Rodrigo, Tania Rodríguez-Santamarta, Verónica Blanco-Lorenzo, Llara Prieto-Fernández, Daniela Corte-Torres, Aitana Vallina, Francisco Domínguez-Iglesias, Saúl Álvarez-Teijeiro, Juana M. García-Pedrero

**Affiliations:** 1Department of Oral and Maxillofacial Surgery, Hospital Universitario Central de Asturias (HUCA), Carretera de Rubín s/n, 33011 Oviedo, Spain; taniasantamarta@gmail.com; 2Instituto de Investigación Sanitaria del Principado de Asturias (ISPA), Instituto Universitario de Oncología del Principado de Asturias (IUOPA), Universidad de Oviedo, Carretera de Rubín s/n, 33011 Oviedo, Spain; palomalequerica@gmail.com (P.L.-F.); jprodrigo@uniovi.es (J.P.R.); prietollara@uniovi.es (L.P.-F.); saul.teijeiro@gmail.com (S.Á.-T.); 3Department of Surgery, University of Oviedo, Julián Clavería s/n, 33006 Oviedo, Spain; 4Department of Biochemistry, Hospital Universitario Central de Asturias (HUCA), Carretera de Rubín s/n, 33011 Oviedo, Spain; 5Department of Otolaryngology, Hospital Universitario Central de Asturias (HUCA), Carretera de Rubín s/n, 33011 Oviedo, Spain; 6Centro de Investigación Biomédica en Red de Cáncer (CIBERONC), Instituto de Salud Carlos III, Av. Monforte de Lemos 3-5, 28029 Madrid, Spain; 7Department of Pathology, Hospital Universitario Central de Asturias (HUCA), Carretera de Rubín s/n, 33011 Oviedo, Spain; veronica.blanco@sespa.es (V.B.-L.); alaicla@hotmail.es (A.V.); 8Principado de Asturias Biobank, Hospital Universitario Central de Asturias (HUCA), Carretera de Rubín s/n, 33011 Oviedo, Spain; mdanielac@hotmail.com; 9Health Research Institute of the Principality of Asturias (ISPA), 33011 Oviedo, Spain; 10Department of Pathology, Hospital Universitario de Cabueñes, Prados 395, 33394 Gijón, Spain; fdoig59@yahoo.es

**Keywords:** oral squamous cell carcinomaLLT1, PD-L1, TILs, prognosis

## Abstract

Lectin-like transcript-1 (LLT1) expression is detected in different cancer types and is involved in immune evasion. The present study investigates the clinical relevance of tumoral and stromal LLT1 expression in oral squamous cell carcinoma (OSCC), and relationships with the immune infiltrate into the tumor immune microenvironment (TIME). Immunohistochemical analysis of LLT1 expression was performed in 124 OSCC specimens, together with PD-L1 expression and the infiltration of CD20^+^, CD4^+^, and CD8^+^ lymphocytes and CD68^+^ and CD163^+^-macrophages. Associations with clinicopathological variables, prognosis, and immune cell densities were further assessed. A total of 41 (33%) OSCC samples showed positive LLT1 staining in tumor cells and 55 (44%) positive LLT1 in tumor-infiltrating lymphocytes (TILs). Patients harboring tumor-intrinsic LLT1 expression exhibited poorer survival, suggesting an immunosuppressive role. Conversely, positive LLT1 expression in TILs was significantly associated with better disease-specific survival, and also an immune-active tumor microenvironment highly infiltrated by CD8^+^ T cells and M1/M2 macrophages. Furthermore, the combination of tumoral and stromal LLT1 was found to distinguish three prognostic categories (favorable, intermediate, and adverse; *p* = 0.029, Log-rank test). Together, these data demonstrate the prognostic relevance of tumoral and stromal LLT1 expression in OSCC, and its potential application to improve prognosis prediction and patient stratification.

## 1. Introduction

Oral squamous cell carcinoma (OSCC) is a highly prevalent malignancy, with five-year survival rates ranging from 45 to 60% [1]. OSCC is an immunosuppressive disease able to evade immune surveillance avoiding an effective immune response [2]. The tumor immune microenvironment (TIME) plays a prominent role in cancer progression and includes various types of immune cells, such as macrophages, lymphocytes, and natural killer (NK) cells. Some of these cells have emerged as promising prognostic biomarkers that combined with the TNM classification could help to refine OSCC prognostic stratification and treatment decision-making. Innate and adaptive immune cells are involved in the response to cancer cells. Most notably, NK cells and cytotoxic CD8^+^ T cells play a paramount role in tumor cell clearance [3]. Other immune cells involved in cancer response are M1 and M2 macrophages and B cells [4]. The programmed cell death protein 1 (PD-1)/PD-ligand 1 protein pathway has emerged as a highly relevant mechanism for tumors to evade the immune system. PD-1 is a receptor primarily expressed on the surface of T cells, but it can also be detected in other immune cells such as NK, B cells, and monocytes [5,6]. In turn, PD-L1 is a ligand expressed by tumor cells. PD-L1 acts as an inhibitory signal of PD-1 [5], thereby reducing T cell function, which ultimately contributes to cancer cells’ evasion of immune detection and destruction [3,6]. Four types of TIMEs have been established according to PD-L1 expression and T cell infiltration [7]: type I (adaptive immune resistance), characterized by positive PD-L1 expression and high CD8^+^ TIL infiltration; type II (immunologic ignorance), showing an absence of PD-L1 expression and low TIL infiltration; type III (intrinsic induction), with positive PD-L1 expression and low TIL density; and type IV (immune tolerance), characterized by a negative PD-L1 expression and high TIL density.

NK cells are innate lymphoid cells that account for 5–20% of the total lymphocytes in peripheral blood [8]. Their prominent role in tumor immunosurveillance is supported by the high cancer incidence observed in those individuals with defective NK functions caused by genetic abnormalities [9,10]. NK cell function resembles that of cytotoxic CD8^+^T cells, except that NK activation does not require antigen-presenting cell priming or MHC restriction [11]. Several inhibitory and activating receptors are expressed on the surface of NK cells, and the balance between these two types of receptors modulates NK cell functions [12]. NK receptors comprise two structurally different classes: the families of immunoglobulin-like receptors and C-type lectin-like receptors (CTLR) [13]. Lectin-like transcript 1 (LLT1), also known as CLEC2D (C-type lectin domain family 2 member D), OCIL (osteoclast inhibitory lectin), and CLAX (lectin-like NK cell receptor), is present in NK cells, and also in T cells, monocytes, macrophages, activated dendritic, and B cells [3]. It is the ligand for CD161 (NKR-P1A or KLRB1) receptor on NK cells [3], upregulated by IL-12 [14]. T cell CD161 expression has been shown to enhance cytokine production, whereas NK cell CD161 receptor inhibits NK cell-mediated cytotoxicity [15]. Beyond immune cells, LLT1 expression could also be detected in other cells and tissues, such as tumor cells to facilitate their escape from NK cell immunosurveillance [3]. LLT1 expression has been found to be upregulated in several cancers such as glioblastoma [16], prostate cancer [15], triple-negative breast cancer [17], and B cell non-Hodgkin’s lymphoma [18]. The expression and clinical significance of LLT1 in OSCC have been scarcely investigated. To date, only one study has been recently published by Hu et al. [19] that examined the expression patterns of CD161/LLT1 in OSCC and the impact on patient prognosis. This prompted us to investigate the clinical relevance of LLT1 expression in the OSCC TIME. To this purpose, LLT1 expression was thoroughly analyzed in both tumor cells and immune cells. Tumoral and stromal LLT1 expression was subsequently correlated with the tumor immune infiltrate, TIME type, clinicopathological variables, and OSCC prognosis.

## 2. Results

### 2.1. Patient Characteristics and Follow-Up

A cohort of 124 OSCC patients was selected for the study. The mean age of all patients was 58.69 years (ranging from 28 to 91); 82 (66%) patients were men, 84 (67.7%) were smokers, and 69 (55.6%) were alcohol drinkers. In total, 49 patients developed lymph node metastasis, 25 (51%) were pN1, and 24 (49%) were pN2. There were no pN3 cases in our series. A total of 20 (16%) tumors were classified as stage I, 32 (25.8%) stage II, 26 (20.9%) stage III, and 47 (37.9%) stage IV. The majority of the tumors (80; 64.5%) were well differentiated, 41 (33%) were moderately differentiated, and only 4 (3.2%) tumors were undifferentiated. All patients were surgically treated with curative intention. None of them underwent chemotherapy or radiotherapy before surgical treatment; however, complementary radiotherapy was administered in 75 (60.5%) and chemotherapy in 14 (11%) cases. From the total cohort of 124 patients, 75 of them were administered adjuvant radiotherapy with radiation doses in the range of 66–70 Gy, delivered in daily fractions of 1.9–2.1 Gy/fraction once daily. Fourteen patients with high-risk features (positive or close margins or extranodal extension) received subsequent adjuvant chemotherapy with regimens that had two- or three-drug combinations, comprising a platinum-based agent (cisplatin or carboplatin) and either paclitaxel/docetaxel with or without 5-fluorouracil. The number of cycles ranged from one to six. Two patients also received cetuximab and one patient nivolumab.

The mean and median follow-up times were 71.82 and 61 months, respectively. At the end of follow-up, 53 (42.4%) patients died of the OSCC. Tumor recurrence was developed in 54 (43.2%) cases. The mean and median follow-up times were 99.23 and 108 months, respectively, for patients without tumor recurrence, while these figures were 35.78 and 18 months, respectively, for patients who suffered a recurrence. The five-year disease-specific survival was 65%.

### 2.2. Immunohistochemical Analysis of LLT1 Expression in OSCC Tissue Specimens

LLT1 staining was detected in the cytoplasm of tumor cells. Tumoral LLT1 expression was valuable in all but one case: 83 (66%) tumors showed negative LLT1 staining (score 0), 37 (29%) showed weak staining (score 1), 4 showed moderate staining (3.2%) (score 2), and none showed strong staining (score 3). Thus, we found a total of 41 (33%) LLT1-positive tumors.

LLT1 expression was also detected in TILs and scored as negative in 69 cases (55%), mild to moderate in 49 (39%) tumors, abundant in 5 (4%), and highly abundant in one case (0.8%). Thus, a total of 55 (44%) tumors showed positive LLT1 staining in TILs. We found a positive significant correlation between stromal and tumoral LLT1 expression (Spearman’s Rho correlation coefficient = 0.301, *p* = 0.001). Representative examples of tumoral and stromal LLT1 immunostaining in OSCC tissue specimens are shown in Figure 1.

### 2.3. Associations of LLT1 Expression with Clinicopathological Variables and Patient Survival

Next, we assess possible associations between both tumoral and stromal LLT1 expression with the clinical and pathological features of OSCC patients. As summarized in Table 1, no significant associations were observed with any of the clinicopathological variables studied.

Kaplan–Meier and univariate Cox analysis revealed that patients harboring LLT1-positive tumors showed a shorter disease-specific survival (92.45 months, standard error—SE = 11.34) than patients with LLT1-negative tumors (139.21 months, SE = 11.88). However, these differences did not reach statistical significance (Log-rank test, *p* = 0.30, HR = 1.40; 95% CI = 0.80–2.46) (Figure 2A). Conversely, patients harboring tumors with LLT1-positive TILs exhibited significantly better disease-specific survival (mean survival of 143.12 months, SE = 14.19) than those with LLT1-negative TILs (mean survival of 120.24 months, SE = 30.11) (Log-rank test, *p* = 0.05, HR = 0.64; 95% CI = 0.36–0.97) (Figure 2B). In addition, we assessed the impact of the combination of tumoral and stromal LLT1 expression on OSCC prognosis. We found that patients with LLT1-negative tumors and LLT1-positive TILs exhibited the highest survival (166.03 months, SE = 17.85), whereas the subgroup of patients with LLT1-positive tumors and LLT1-negative TILs had the worst prognosis (76.48 months, SE = 20.01) (Figure 2C). The two remaining subgroups showed intermediate survival rates: patients with LLT1-negative tumors and LLT1-negative TILs (124.37 months, SE = 14.58), and patients with LLT1-positive tumors and LLT1-positive TILs (93.75 months, SE = 12.29) (Figure 2C). By combining these two intermediate subgroups, we were able to improve the prognosis prediction of tumoral and stromal LLT1 (Log-rank test, *p* = 0.029) (Figure 2D) and to distinguish three prognostic categories: favorable (i.e., tumor-negative/stroma-positive, 166.03 months, SE = 17.85), intermediate (i.e., tumor-positive/stroma-positive and tumor-negative/stroma-negative, 123.42 months, SE = 12.23), and adverse (i.e., tumor-positive/stroma-negative, 76.48 months, SE = 20.01) (Figure 2D). Taking the intermediate category as a reference, the favorable category was associated with a better outcome (HR = 0.47; 95% CI = 0.21–1.06), and the adverse category to a poorer outcome (HR = 2.22; 95% CI = 1.01–4.96).

Patients with T3–T4 tumors, as well as patients with neck lymph node metastasis, and III and IV clinical stages, exhibited significantly worse disease-specific survival (*p* = 0.001, *p* = 0.01, and *p* = 0.002; respectively). Although patients with well-differentiated tumors showed better survival, no significant differences in survival rates were observed among the different grades of tumor differentiation. Multivariable Cox regression analysis, including T classification (T1–T2 vs. T3–T4), N classification (N0 vs. N+), and LLT1 expression in TILs showed that the parameters independently associated with worse disease-specific survival were T3–T4 classification (HR = 2.55; 95% CI = 1.47–4.42; *p* = 0.001) and neck lymph node metastasis (HR = 1.89; 95%CI = 1.10–3.25; *p* = 0.02). However, LLT1 expression in TILs did not retain its significant association with a better prognosis (*p* = 0.09).

The potential impact of LLT1 on the risk of recurrence was also assessed using the Kaplan–Meier analysis. Patients harboring LLT1-positive tumors showed a lower mean disease-free survival (DFS) of 92.98 (95% CI = 70.59–115.37) than those with LLT1-negative tumors (mean DFS of 136.93; 95% CI = 113.35–160.51). However, differences were not statistically significant (Log-rank test, *p* = 0.26). On the other hand, patients harboring tumors with LLT1-positive TILs exhibited a slightly higher mean DFS of 129.89 (95% CI = 103.18–156.60) compared to those with LLT1-negative TILs (mean DFS of 124.28; 95% CI = 96.53–152.03; Log-rank test, *p* = 0.59). Therefore, LLT1 expression was not found to be a recurrence risk predictor in this OSCC cohort. Nevertheless, these data reinforce our findings regarding the prognostic relevance of tumoral and stromal LLT1 on OSCC patient survival, further supporting a relationship of tumoral LLT1 with poor survival rates and a possible immunosuppressive role, whereas stromal LLT1 expression in TILs is related to favorable outcomes in OSCC and an immune-active highly infiltrated TIME.

### 2.4. Relationships between Tumoral and Stromal LLT1 Expression and the Tumor Immune Microenvironment

We explored possible associations of tumoral LLT1 expression and the density of distinct infiltrating immune cell populations, such as CD8^+^ T, CD4^+^ T, and CD20^+^ B TILs and CD68^+^ and CD163^+^ macrophage subtypes (Table 2). Tumoral LLT1 expression was significantly associated with the infiltration of CD8^+^ T cells in both the tumor nests (intratumoral) and surrounding stroma. The mean number of stroma- and tumor-infiltrating CD8^+^ T cells was higher in LLT1-positive tumors (Kruskal–Wallis test, *p* = 0.03 and *p* = 0.003, respectively). Similarly, the mean number of tumor-infiltrating CD20^+^ B cells was higher in LLT1-positive tumors compared to LLT1-negative tumors (Kruskal–Wallis test, *p* = 0.002). There was also a tendency for higher mean numbers of stroma- and tumor-infiltrating CD4^+^ T cells and stroma-infiltrating CD20^+^ B cells in LLT1-positive tumors, although these differences did not reach statistical significance.

A similar trend was observed when tumoral LLT1 expression was correlated with macrophage infiltration. The mean numbers of CD68^+^ and CD163^+^ infiltrating macrophages in both the tumor nests and stroma were concordantly higher in LLT1-positive tumors compared to LLT1-negative tumors. Differences were statistically significant between tumoral LLT1 expression and stroma- and tumor-infiltrating CD68^+^ macrophages (Kruskal–Wallis test, *p* = 0.03 and *p* = 0.001, respectively), and also tumor-infiltrating CD163^+^ macrophages (Kruskal–Wallis test, *p* = 0.002).

Concordantly, the mean densities of all these immune cell populations were also higher in tumors harboring LLT1-positive TILs compared to those with LLT1-negative TILs. However, differences were only statistically significant for the intratumoral infiltration of CD8^+^ T cells, CD68^+,^ and CD163^+^ macrophages (Table 2).

Furthermore, we also evaluated the relationship between LLT1 expression and the four types of immune microenvironments described by Teng et al. [7]. Following this classification, the frequencies of TIME types in our series were as follows: type I (positive PD-L1 expression and high density of CD8^+^ TILs), 13 (10%) cases; type II (negative PD-L1 expression and low CD8^+^ TIL density), 40 (32%) cases; type III (positive PD-L1 expression and low CD8^+^ TIL density), 5 (4%) cases; and type IV (negative PD-L1 expression and high CD8^+^ TIL density), 64 (51%) cases.

Tumoral LLT1 expression was found to significantly correlate with the type of TIME (Fisher exact test, *p* = 0.018) (Table 3). LLT1-positive tumors were more frequently associated with a type I TIME (adaptive immune resistance, 54%), followed by type IV (immune tolerance, 36%). The two remaining TIME types related to low densities of CD8^+^ TILs (i.e., type II, immunological ignorance and type III, intrinsic induction) were rare in LLT1-positive tumors (20% type II, and 0% type III) (Table 3). Regarding stromal LLT1 expression, LLT1-positive TILs were more frequently observed in those tumors with a type I TIME (77%) followed by type IV (42%), type II (39%), and type III (20%), with a borderline-significant association (Fisher exact test, *p* = 0.06) (Table 3).

## 3. Discussion

After years of research, the complex/heterocellular TIME has emerged as a marker of sensitivity to immunotherapy and a predictor of outcome [20]. Based on TIME composition, tumors have been classified into three different immune profiles [20]: “T cell cold” tumors or “immune desert”, showing no lymphocyte infiltration in either tumor nests or within the TIME as a whole; “immune-excluded” tumors, characterized by an intense infiltrate of immune cells in the stroma surrounding tumors nests, but without lymphocyte infiltration into the tumor nests themselves; and finally, “T cell hot”, “inflamed” or “immune-active” tumors, defined by the lymphocytic infiltration of tumor nests, with the immune cells in close proximity to tumor cells [21,22]. Kather et al. [23] performed an analysis of lymphoid and myeloid phenotypes of human solid tumors and found that immune-excluded tumors were common in head and neck cancers.

Accordingly, Teng et al. [7] distinguished four different types of TIMEs: type I is characterized by a high infiltration of TILs and positive PD-L1 expression driving adaptive immune resistance and is likely to benefit from PD-1/PD-L1 blockade immunotherapy; type II harbors negative PD-L1 expression and negative TIL infiltration, indicative of immunologic ignorance; type III shows positive PD-L1 expression and an absence of TILs indicating intrinsic induction by oncogenic pathways; and type IV, characterized by a negative PD-L1 expression and a high density of TILs, indicating immune tolerance (presumably due to other immune suppressors beyond the PD-1/PD-L1 axis). Here, we found that the most frequent types of TIMEs were type IV (51%) followed by type II (32%), which could reflect the degree of mutagenesis in our series [24]. In type IV TIME, there is a substantial number of M2 macrophages that can be switched to the M1 phenotype in order to reduce the tumor growth and also high TILs; however, the PD-1/PD-L1 axis cannot be targeted and other immunosuppressive pathways not yet well known should be addressed in the future [7]. In turn, type II lacks a detectable immune reaction and still requires designing a therapy to bring T cells into tumors to be considered in future therapies. In this study, LLT1 expression was evaluated in both tumor cells and TILs in an OSCC cohort of 124 patients and associated most frequently with type I, followed by types IV and II. Type I tumors are the most likely to benefit from single-agent anti-PD-1/PD-L1 therapy. Nevertheless, it is also worth taking into account that PD-L1 expression changes dynamically during tumor progression [25], and consequently, PD-L1 expression analysis in archival tissue specimens may not adequately reflect tumor-immune complexity. PD-L1 assessment in pretreatment biopsies might be a more informative approach to accurately evaluate the state of TIME for immunotherapy.

NK cells are able to recognize cells with downregulated MHC-I complexes, while healthy cells send inhibitory signals to NK cells [3]. In the TIME context, LLT1/CD161 complex interaction plays important roles in tumor development and the modulation of immune response. LLT1 appears to have a dual function dependent on the immune cell where it is expressed [3]. NK cells store several contents including granulysin, perforin, and granzyme [3], which are exocytosed to target cells. This leads to the activation of Fas- and TRAIL-associated apoptosis [26,27], ultimately causing effector cell death. LLT1 can be found expressed in different cell types and tissues, including OSCC specimens and surrounding infiltrating NK cells, as shown in this study. The possibility exists for cancer cells to escape from NK cell-mediated immune surveillance [28]. LLT1-mediated tumor evasion could be due to impaired anti-tumor NK effector mechanisms. Based on the role of LLT1 in tumor progression, it has been considered that using LLT1 blocking antibodies could plausibly be a useful strategy to enhance NK cell cytotoxicity, leading to cell lysis, and a novel treatment to prevent tumor metastasis [15,17,26]. In addition, antibody-dependent cell-mediated cytotoxicity is another function of NK cells [26].

Tumoral LLT1 expression has been reported in non-Hodgkin’s lymphomas [18], nodular lymphocyte-predominant Hodgkin lymphomas [29], gliomas [16], prostate cancer [15], triple-negative breast cancer [17], colon cancer [30], lung cancer [31], cutaneous squamous cell carcinomas [32], and in HPV-negative oropharyngeal squamous cell carcinomas [33]. Studies have demonstrated that high densities of immune cells within the TIME are associated with increased patient survival [34]. In solid tumors, T lymphocytes and macrophages are among the most abundant immune cells and are well known as good predictors of clinical outcomes [23]. On the other hand, NK cell infiltration has been associated with improved prognoses in breast cancer, gastric cancer, head and neck cancers, lung cancer, melanoma, neuroblastoma, and hematologic malignancies [35]. This study provides original evidence for the prognostic relevance of tumoral and stromal LLT1 in OSCC. Patients harboring positive tumoral LLT1 expression exhibited poor survival, suggesting a possible immunosuppressive effect of tumor-intrinsic LLT1 expression. In marked contrast, LLT1 expression in TILs was significantly associated with better survival, which could be due to cytokine secretion by T cells. We also found that stromal LLT1 expression was significantly associated with the intratumoral infiltration of CD8^+^ T cells and macrophages, hence inflamed tumors. Therefore, the association of stromal LLT1 expression with a better prognosis may reflect a more immune-active tumor microenvironment highly infiltrated by multiple immune cell populations (CD8^+^TILs, M1/M2 macrophages, and LLT1-positive NK cells). These results are in line with our previous observations in head and neck cutaneous squamous cell carcinomas [32] and HPV-negative oropharyngeal squamous cell carcinomas [33]. A recent study by Hu et al. [19] reported that fewer CD8^+^ T cells were present in those OSCC patients harboring high LLT1 expression in tumor cells (both at the front and the center of tumor tissue), although these differences were not statistically significant. Moreover, these authors also found that tumors with high LLT1+ lymphocyte infiltration showed significantly higher absolute counts of CD3^+^ and CD4^+^ T cells. These results were consistent with a previous study on oropharyngeal squamous cell carcinoma [33].

In our study, LLT1 expression was higher in TILs than in tumor cells, and tumoral and stromal LLT1 were both significantly associated with a higher density of CD8^+^ T cells within the tumor islands. Our findings are fully concordant with the above-referred previous study on oropharyngeal squamous cell carcinomas [33].

The partial inconsistencies between our findings and those by Hu et al. [19] could be due to differences in the activation state of cells [34], different antibodies [28], methodologies/cut-off points used, and differences between the patient cohorts studied in terms of risk factors or clinical and biological characteristics.

The molecular mechanisms underlying LLT1 expression regulation are as of yet poorly understood, and the detection of gene transcripts does not guarantee protein expression [35,36]. LLT1 upregulation is triggered by T-helper cells [31], and LLT1 stimulation appears to be involved in the activation of B lymphocyte responses [29]. LLT1 upregulation has been associated with the cellular activation status [37,38]. Thus, LLT1 expression has been detected in stimulated T, B, and NK cells, whereas LLT1 was not observed in circulating monocytes or immature monocyte-derived dendritic cells (DC). Moreover, LLT1 is upregulated on Toll-like receptor (TLR)-activated mature monocytes [37], with IFN-γ being a key signal amplifying LLT1 induction by TLR on antigen-presenting cells (APCs) [39]. LLT1/CD161 is a ligand/receptor pair that regulates both innate and adaptive immune responses [39], and the expression of LLT1 and CD161 in CD68^+^ macrophages and CD4^+^ T cells may allow for intercellular (antigen-presenting cell–T cell) communication [40]. LLT1 expression has also been shown in joint macrophages of rheumatoid arthritis patients [40].

In this study, we found a significant relationship between tumoral and stromal LLT1 expression and the presence of tumor-infiltrating CD163^+^ M2 macrophages (i.e., protumoral and immunosuppressive phenotype), suggesting that LLT1 might have an immunosuppressive role in OSCC. Concerning the inhibitory function of LLT1 in NK cells, it has been pointed out that LLT1 expression by malignant glioma cells could be a mechanism of immune escape preventing the elimination of tumor cells by innate immune responses [16]. Our findings support the presence of different TIL subtypes in the OSCC TIME interacting with one another to exert their effects, thereby highlighting the idea that the intricate interplay between cancer cells and host immune cells plays an important and complex role in the progression of this disease. The differential function of LLT1 in NK and T cells could indicate a certain plasticity in signaling pathways leading to different outcomes depending on cell type, cellular activation status, the stage of development, and the type of stimulation [37,38,41]. LLT1 has been described as a multi-functional protein [42]. The rapid upregulation of LLT1 by distinct immune cell types including T cells, B cells, and NK cells suggests a potential universal role as an early activation marker [38]; however, its role in T cells remains controversial [39]. Furthermore, growing evidence has been provided for LLT1 expression upregulation by non-hematopoietic cells (e.g., epithelial cells) in response to pro-inflammatory cytokines such as IL-1β, TNF-α, or type I interferons [43]. Bambard et al. [44] reported that LLT1 is an activator of IFN-γ production in NK cells, with IFN-γ being the major cytokine produced by NK cells upon the detection of cancerous cells [45], and that LLT1 downstream signaling is likely dependent upon Src-protein tyrosine kinase (Src-PTK), p38 and ERK signaling pathways [44]. IFN-γ is not stored by cells but secreted immediately after synthesis [46], which suggests that LLT1 stimulates IFN-γ production by modulating post-transcriptional or translational events [44].

Regarding the prognostic significance of LLT1 expression, our findings are in line with those recently reported by Hu et al. [19], consistently revealing that tumor-intrinsic LLT1 expression was associated with poor clinical outcomes in OSCC, whereas positive LLT1 expression in TILs was associated with a favorable prognosis. The study by Hu et al. [19] was performed in a Chinese cohort of 109 OSCC patients with several demographic, etiological, and clinicopathological differences from our studied Spanish patient cohort. The Hu series is composed of 63% of patients older than 60 years, 50% with positive neck lymph node metastasis, and 78% of their tumors were moderately or poorly differentiated. In our series, 61% of patients were younger than 65 years, 39% of cases presented neck lymph node metastasis (63% of them were pN1), and well-differentiated tumors were predominant (64% of cases). Despite all the differences between the two studied OSCC cohorts and others such as the antibodies used and the immunohistochemical evaluation, both studies led to consistent results in terms of the impact of tumoral and stromal LLT1 on OSCC prognosis and outcomes.

Interestingly, we also found that the combination of tumoral and stromal LLT1 was valuable to stratify OSCC patients into three prognostic categories: (1) favorable, patients with LLT1-negative tumors and LLT1-positive TILs; (2) intermediate, patients with either LLT1-positive tumors and LLT1-positive or LLT1-negative tumors and LLT1-negative TILs; and (3) adverse prognosis, those patients harboring LLT1-positive tumors and LLT1-negative TILs. Altogether, these data demonstrate a potential application for tumoral and stromal LLT1 expression as a prognostic predictor to improve OSCC patient stratification. Our results also provide the rationale for designing novel strategies aimed at specifically targeting tumoral LLT1 expression, which is related to the worst prognostic subgroup.

We are aware that our study has some limitations. This is a retrospective study in nature, which may bias the results. The analysis of tumor tissue with immunohistochemistry is the gold standard to assess the amount and spatial distribution of cells in the tumor immune infiltrate allowing for the exact quantification of the type, density, and spatial distribution of immune cells [23]. Nevertheless, we have used tissue microarrays with a limited amount of tumor and stroma tissue; however, it is worth mentioning that the expression patterns were quite concordant among the three tissue cores selected for analysis from each tumor block.

## 4. Materials and Methods

### 4.1. Patients and Tissue Specimens

Formalin-fixed paraffin-embedded (FFPE) tissue specimens from 124 patients with histologically confirmed OSCC were retrospectively collected, in accordance with approved institutional review board guidelines. Patients underwent surgical treatment for curative purposes at the Hospital Universitario Central de Asturias (HUCA) between 1 March 2000 and 31 December 2010. All experimental procedures were conducted in accordance with the Declaration of Helsinki and approved by the Institutional Ethics Committee of the HUCA and by the Regional Ethics Committee from Principado de Asturias (date of approval 14 May 2019; approval number 136/19, for the project PI19/01255). Inclusion criteria in this study were an OSCC diagnosis and radical resection of the primary tumor with simultaneous neck lymph node dissection. The disease stage was determined according to the 8th edition of the AJCC Cancer Staging Manual [47]. Clinicopathological data were collected from medical records, and tissue specimens were provided by the Principado de Asturias BioBank, part of the Spanish National Biobanks Network (PT20/00161 and PT23/00077). Representative tissue areas from the 124 OSCC patients were obtained from archival FFPE tumor blocks to construct tissue microarrays (TMAs), as previously described [48].

### 4.2. Immunohistochemistry

The OSCC TMAs were cut into 3 μm sections and dried on Flex IHC microscope slides (DakoCytomation, Glostrup, Denmark). Sections were deparaffinized with standard xylene and hydrated through graded alcohols into water. Antigen retrieval was performed by heating the sections with the Envision Flex Target Retrieval solution (Dako). The staining was performed at room temperature on an automatic staining workstation (Dako Autostainer Plus, Dako) The staining was conducted at room temperature on an automatic staining workstation (Dako Autostainer Plus, Dako) with mouse monoclonal antibody anti-LLT1/CLEC2D (Clone 4C7 #H00029121-M01; Novus Biologicals, Littleton, CO, USA) at 1:200 dilution; also, monoclonal antibodies against CD20 (Dako, clone L26, #M0755; 1:200 dilution), CD4 (Dako, clone 4B12, 1:80 dilution), CD8 (Dako, clone C8/144B, prediluted), PD-L1 antibody (PD-L1 IHC 22C3 pharmDx, Dako SK006, clone 22C3, 1:200 dilution), CD68 (Agilent-Dako, clone KP1, prediluted), and CD163 (Biocare Medical, Pacheco, CA, USA, clone 10D6, 1:100 dilution). The antibody–antigen complexes were visualized using the Dako EnVision Flex + Visualization System (Dako) and diaminobenzidine chromogen as substrate.

LLT1 immunostaining was preferentially detected in the cytoplasm of tumor cells, although some cases showed protein enrichment at the cell membrane. Expression levels were independently scored by two observers blinded to clinical information. A scoring system based on staining intensity was applied, and LLT1 expression was classified as negative (score 0), weak (score 1), moderate (score 2), or strong staining (score 3). There was a high inter-observer agreement (>95%). For statistical purposes, staining scores were dichotomized as negative expression (score 0) versus positive expression (scores 1 to 3). There were no disagreements between the two observers when the expressions were finally dichotomized.

LLT1 staining in TILs was jointly evaluated in the intratumoral compartment and the stromal compartment and the densities of positive immune cells were scored as follows: no or sporadic positive cells (score 0); mild to moderate numbers of positive cells (score 1); abundant positive cells (score 2); and highly abundant positive cells (score 3). Finally, scores were dichotomized as negative (score 0) versus positive expression (scores 1 to 3). There were no inter-observer disagreements.

The number of CD20^+^, CD68^+^, CD163^+^, CD4^+^, CD8^+,^ and FOXP3^+^ cells in both the tumor nests and the tumor stroma was counted in each 1 mm^2^ area from three independent high-powered representative microscopic fields (HPFs, 400×). PD-L1 expression in more than 10% of tumor cells was previously found to be significantly associated with poorer survival [49], and, therefore, established as a cut-off point for subsequent analyses.

### 4.3. Statistical Analysis

The statistical analysis was performed using IBM SPSS for Windows (version 27.0.1, IBM-SPSS Inc., Armonk, NY, USA). The clinicopathological characteristics of patients were presented as absolute frequencies, percentages, means, and medians. Bivariate analysis to assess associations between variables was examined using χ^2^ or Fisher’s exact test, Spearman’s test, or Mann–Whitney’s test, as appropriate. Disease-specific survival (DSS) was determined from the date of treatment completion to death from the tumor. Patients who were alive at the time of the last follow-up and those who died of causes other than cancer were censored. DSS was estimated using the Kaplan–Meier method and compared with the Log-rank test. Hazard ratios (HR) with their 95% confidence intervals (CI) were calculated using the univariate Cox proportional hazards model. All tests were two-sided and *p*-values equal to or less than 0.05 were considered statistically significant.

## 5. Conclusions

This study demonstrates the prognostic relevance of tumoral and stromal LLT1 expression in OSCC, and its potential application to improve prognostic predictions and patient stratification. Notably, the combination of tumoral and stromal LLT1 was found to distinguish three distinct prognostic categories (i.e., favorable, intermediate, and adverse prognosis in those patients harboring LLT1-positive tumors and LLT1-negative TILs). Our results also provide the rationale for designing novel strategies aimed at specifically targeting tumoral LLT1 expression, which is related to the worst prognostic subgroup. Therefore, these findings could serve to implement novel prognostic biomarkers and therapeutic innovations for OSCC patients.

## Figures and Tables

**Figure 1 ijms-25-04314-f001:**
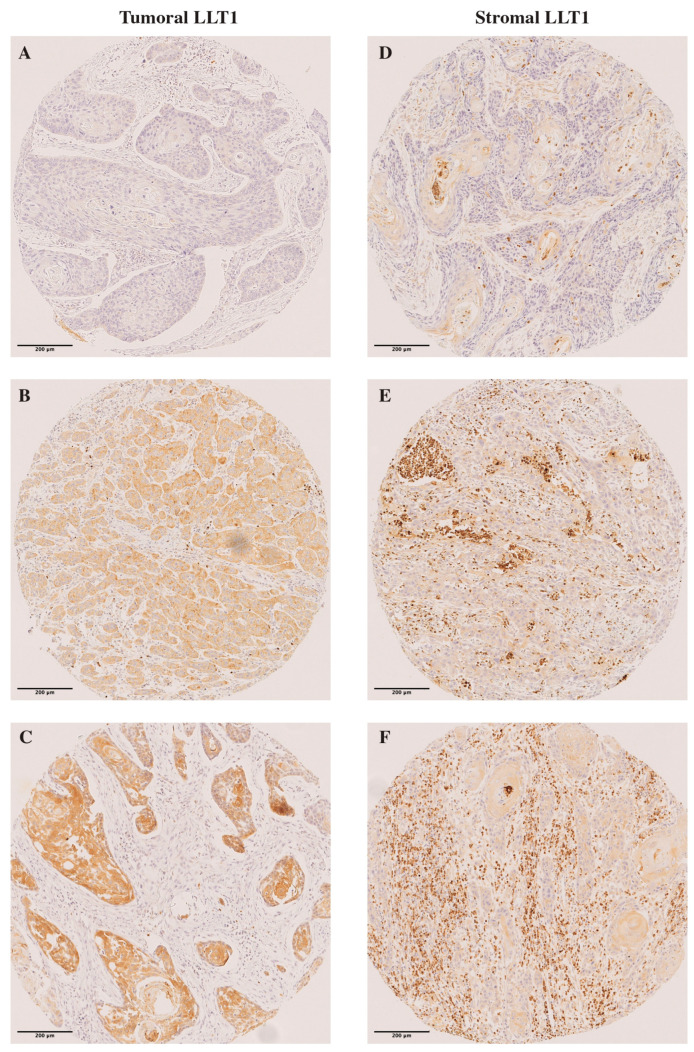
Immunohistochemical analysis of LLT1 in OSCC tissue specimens. Representative images of tumors showing (**A**) negative LLT1 expression (score = 0); two examples of tumoral LLT1 expression showing (**B**) weak (score = 1) and (**C**) moderate staining (score = 2); three examples of stromal LLT1 expression showing (**D**) mild to moderate positive TILs (score = 1); (**E**) abundant (score = 2); and (**F**) highly abundant positive TILs (score = 3). Scale bar, 200 µm. Magnification, 20×.

**Figure 2 ijms-25-04314-f002:**
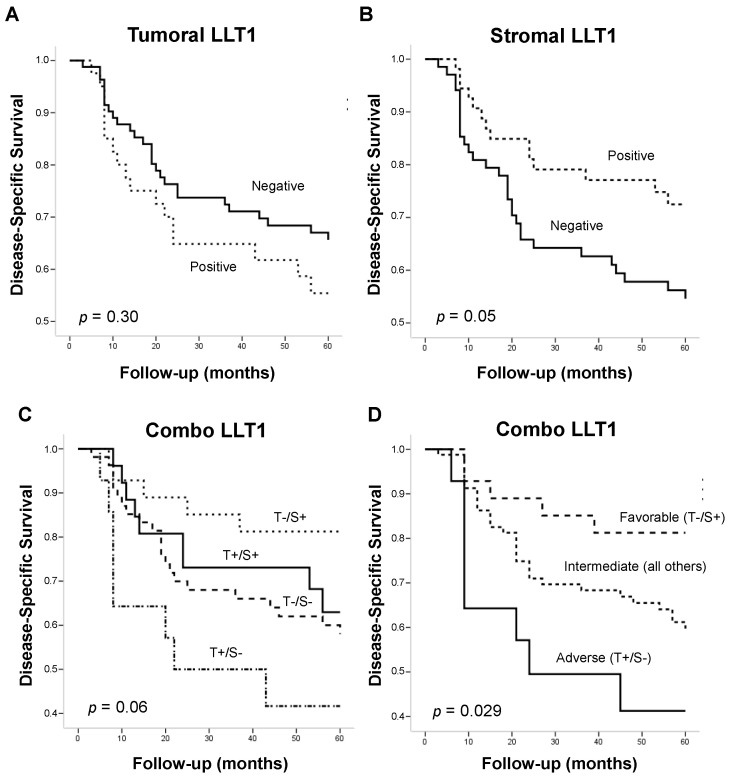
Kaplan–Meier disease-specific survival in the cohort of 125 OSCC patients categorized by (**A**) tumoral LLT1 expression (positive versus negative), (**B**) stromal LLT1 expression in TILs (positive versus negative), and (**C**) combination of tumoral and stromal LLT1 expression into either four or (**D**) three prognostic subgroups (favorable, intermediate, and adverse). *p*-values were estimated with the Log-rank test.

**Table 1 ijms-25-04314-t001:** Associations between tumoral and stromal LLT1 expression and the clinicopathological characteristics of 124 OSCC patients studied.

Variable	No. Cases	Tumoral LLT1-Positive Expression (%)	*p*	Stromal LLT1-Positive Expression (%)	*p*
**Age**					
<65 years	76	23 (30)	0.40 ^#^	35 (46)	0.63 ^#^
≥65 years	48	18 (37)		20 (42)	
**Gender**					
Men	81	27 (33)	0.93 ^#^	33 (41)	0.26 ^#^
Women	43	14 (33)		22 (51)	
**Tobacco use**					
Smoker	83	26 (31)	0.55 ^#^	38 (46)	0.64 ^#^
Non-smoker	41	15 (37)		17 (42)	
**Alcohol use**					
Drinker	68	22 (32)	0.85 ^#^	27 (40)	0.25 ^#^
Non-drinker	56	19 (34)		28 (50)	
**Tumor status**					
pT1	27	8 (30)	0.91 *	9 (33)	0.18 *
pT2	54	17 (32)		28 (52)	
pT3	16	6 (38)		9 (56)	
pT4	27	10 (37)		9 (33)	
**Nodal status**					
pN0	76	24 (32).	0.38 *	35 (46)	0.33 *
pN1	25	11 (44).		8 (32)	
pN2	23	6(26)		12 (52)	
**Clinical stage**					
Stage I	20	6 (30)	0.85 *	7 (35)	0.19 *
Stage II	32	9 (28)		19 (59)	
Stage III	26	9 (35)		9 (35)	
Stage IV	46	17 (37)		20 (44)	
**WHO histopathological grade**					
G1	79	24 (30)	0.40 ^#^	34 (43)	0.69 ^#^
G2–G3	45	17 (38)		21 (47)	
**Follow-up**					
Alive	52	16 (31)	0.59 *	27 (52)	0.24 *
Dead by the disease	53	20 (38)		19 (36)	
Censored	19	5 (26)		9 (47)	

*p*-value calculated using * Chi squared, ^#^ Fisher’s exact test.

**Table 2 ijms-25-04314-t002:** Associations between tumoral and stromal LTT1 expression and the infiltration of CD4^+^, CD8^+^, and CD20^+^ TILs and CD68^+^ and CD163^+^ macrophages in 124 OSCC patients.

Factor (Mean, SD)	Tumoral LLT1 Expression		*p* *	Stromal LLT1 Expression		*p* *
	Negative	Positive		Negative	Positive	
**Stroma-infiltrating CD4^+^**	46.17 (49.57)	70.52 (95.65)	0.08	53.12 (77.60)	57.34 (55.04)	0.42
**Tumor-infiltrating CD4^+^**	5.41 (11.84)	7.51 (13.45)	0.11	4.66 (10.40)	7.82 (14.27)	0.08
**Stroma-infiltrating CD8^+^**	148.09 (154.81)	240.89 (274.72)	0.03	172.91 (201.49)	188.43 (207.50)	0.49
**Tumor-infiltrating CD8^+^**	41.25 (59.29)	62.45 (51.50)	0.003	34.93 (46.72)	64.53 (64.94)	0.005
**Stroma-infiltrating CD20^+^**	33.68 (62.80)	60.57 (104.09)	0.08	37.68 (73.34)	49.25 (85.52)	0.09
**Tumor-infiltrating CD20^+^**	1.31 (3.11)	2.32 (3.76)	0.002	1.34 (3.12)	2.01 (3.62)	0.35
**Stroma-infiltrating CD68^+^**	111.48 (69.95)	148.35 (101.46)	0.03	111.27 (72.73)	139.15 (91.03)	0.11
**Tumor-infiltrating CD68^+^**	42.17 (37.92)	70.94 (52.43)	0.001	37.36 (36.68)	68.65 (48.93)	<0.0001
**Stroma-infiltrating CD163^+^**	162.78 (95.35)	179.53 (102.48)	0.48	166.63 (91.06)	172.34 (105.40)	0.92
**Tumor-infiltrating CD163^+^**	25.20 (22.61)	43.66 (36.33)	0.002	24.59 (27.25)	39.66 (29.13)	0.001

* U Mann–Whitney.

**Table 3 ijms-25-04314-t003:** Associations between tumoral and stromal LLT1 expression and the type of tumor immune microenvironment (TIME).

Type of TIME	Tumoral LLT1 Expression		*p*	Stromal LLT1 Expression		*p*
	Negative	Positive		Negative	Positive	
**Type I (PD-L1+/CD8^+^ high)**	5 (40%)	8 (62%)		3 (23%)	10 (77%)	
**Type II (PD-L1−/CD8^+^ low)**	31 (80%)	8 (21%)	0.018	24 (62%)	15 (39%)	0.06
**Type III (PD-L1+/CD8^+^ low)**	5 (100%)	0 (0%)		4 (80%)	1 (20%)	
**Type IV (PD-L1−/CD8^+^ high)**	41 (64%)	23 (36%)		37 (58%)	27 (42%)	

*p*-value calculated using the Fisher’s exact test.

## Data Availability

The data are available upon request to the corresponding author (J.C.d.V.) due to privacy/ethical restrictions.

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
