# Peer review of "Lectin-like Transcript-1 (LLT1) Expression in Oral Squamous Cell Carcinomas: Prognostic Significance and Relationship with the Tumor Immune Microenvironment"

_ijms, 2024, doi:10.3390/ijms25084314_

Round 1
Reviewer 1 Report
Comments and Suggestions for Authors
In this study, the authors investigate the expression of Lectin-like transcript-1 (LLT1) expression in oral squamous cell 2 carcinomas (OSCC). The authors successfully demonstrated the prognostic relevance of tumoral and stromal LLT1 expression in OSCC. Although the findings would be important in clinical setting in terms of understanding the molecular pathogenesis of OSCC, another paper has recently shown a similar result. Therefore, the author should revise the manuscript to emphasize the novelty of the study.
Major comment
1. Hu et al have recently reported that the high expression of tumor cells-derived LLT1 confers poor clinical outcomes in OSCC. They also showed that higher CD161+ and LLT1+ in tumor-infiltrated lymphocytes are associated with better prognosis. Therefore, the authors must refer the paper (PMID: 38502058), and discuss the similarities and differences of the study compared with the published paper.
2. The authors should discuss the reason why the mean number of stroma- and tumor-infiltrating CD8+ T cells and macrophages are higher in LLT1-positive tumors.
3. How about the relationship between LLT1-positive tumors and metastasis in this study?
4. How about the relationship between LLT1-positive tumors and cell proliferation marker expression (e.g. Ki-67)?
5. How about the relationship between LLT1-positive tumors and innate immune activation (e.g. interferons expression)?
Author Response
REVIEWER # 1
Comments and Suggestions for Authors
In this study, the authors investigate the expression of Lectin-like transcript-1 (LLT1) expression in oral squamous cell 2 carcinomas (OSCC). The authors successfully demonstrated the prognostic relevance of tumoral and stromal LLT1 expression in OSCC. Although the findings would be important in clinical setting in terms of understanding the molecular pathogenesis of OSCC, another paper has recently shown a similar result. Therefore, the author should revise the manuscript to emphasize the novelty of the study.
Major comment
- Hu et al have recently reported that the high expression of tumor cells-derived LLT1 confers poor clinical outcomes in OSCC. They also showed that higher CD161+ and LLT1+ in tumor-infiltrated lymphocytes are associated with better prognosis. Therefore, the authors must refer the paper (PMID: 38502058), and discuss the similarities and differences of the study compared with the published paper.
Response: The paper by Hu et al. has now been included in this new revised version of the manuscript. Differences and similarities between both studies are now discussed, as recommended.
- The authors should discuss the reason why the mean number of stroma- and tumor-infiltrating CD8+ T cells and macrophages are higher in LLT1-positive tumors.
Response: In the study by Hu et al. reported fewer CD8+ T cells in the group of tumors with high LLT1 expression (both at the tumor front and the center of tumor tissue); however, these differences were not statistically significant (Figure 3B and C). In addition, these authors also described: “Unexpectedly, we observed that tumors with increased LLT1+ lymphocyte infiltration had a higher absolute count of CD3+ T cells as well as CD3+ CD4+ T cells (p = 0.0002 and p = 0.0118; Figure 4B), which was consistent with a previous study on oropharyngeal squamous cell carcinoma [23]”.
In our study, LLT1 expression was higher in TILs than in tumor cells, and tumoral and stromal LLT1 were both significantly associated with higher density of CD8+ T cells within the tumor islands. Our results are consistent with the above-referred previous study on oropharyngeal carcinomas (Sánchez-Canteli et al Biomedicines 2020, now ref. 33). In addition, we found a significant relationship between tumoral and stromal LLT1 expression and the presence of tumor-infiltrating CD163+ M2 macrophages (i.e. protumoral and immunosuppressive phenotype), suggesting that LLT1 might have an immunosuppressive role in OSCC.
Differences between our findings and those by Hu et al. could be due to differences in the activation state of cells, different antibodies/methodologies/cut-off points used as well as differences between the patient cohorts studied in terms of risk factors, clinical and biological characteristics.
- How about the relationship between LLT1-positive tumors and metastasis in this study?
Response: Tumoral LLT1 expression was found in 58% N0 cases and in 42% N+ cases with positive lymph node metastasis. 58% N0 cases showed LLT1 expression in tumor cells and 64% LLT1-positive expression in TILs, whereas LLT1 expression was negative in 42% and 36%, respectively. These differences were not statistically significant. Therefore, there were no association between LLT1 expression and lymph node metastasis in our OSCC cohort.
On the other hand, please note that new data have been included in our revised version of the manuscript to assess the impact of clinicopathological variables as well as LLT1 expression in both univariable and multivariable analyses, as requested by Reviewer # 2 (please see our response to point 1). Results from these analyses showed that lymph node metastasis and T classification were significant predictors of DSS in both univariable and multivariable analyses, whereas LLT1 expression in TILs did not retain a significant association with a better prognosis in multivariable Cox analysis (p = 0.09).
It is quite surprising that in the study by Hu et al. LLT1 expression was found a significant independent prognostic factor in the multivariable Cox analysis but not lymph node metastasis. It has been extensively demonstrated and well accepted that the presence of lymph nodal metastasis is the most important pathologic factor associated with adverse prognosis in OSCC patients and widely used as a valuable tool to guide treatment decision-making. Therefore, these data indicate that our selected OSCC sample is representative of the expected clinical characteristics and behavior, which is fundamental to avoid biases related to non-representative cohorts or unbalanced studies.
- How about the relationship between LLT1-positive tumors and cell proliferation marker expression (e.g. Ki-67)?
Response: Our study did not assess specifically the relationship between LLT1 expression and cell proliferation. Nevertheless, we did not find a significant association with the tumor size (i.e. pT classification) as a related estimate of tumor cell growth.
- How about the relationship between LLT1-positive tumors and innate immune activation (e.g. interferons expression)?
Response: In fact, LLT1 is an activator of IFN-γ production on NK cells, being IFN-γ the major cytokine produced by NK cells upon detection of cancerous cells. LLT1 downstream signaling is likely dependent on Src-protein tyrosine kinase (Src-PTK), p38 and ERK signaling pathways. IFN-γ is not stored by cells but secreted immediately after synthesis, which suggest that LLT1 stimulates IFN-γ production by modulating post-transcriptional or translational events. This information has now been included in the manuscript.
Reviewer 2 Report
Comments and Suggestions for Authors
It was a pleasure to read this interesting and well-conducted study exploring the prognostic impact of LLT1 expression in OSCC. I have one major comment and other minor ones in order to improve the manuscript.
1. The lack of multivariate analysis is the major issue of this study. KM curves and univariate analysis are shallow for today’s standard. LLT1 was not associated with any of the clinicopathological parameters of the tumors, but the impact of them on survival was not assessed. On the other hand, LLT1 was significantly associated with the infiltration of some immune cells. Several studies have demonstrated that the composition of the inflammatory infiltrate is significantly associated with survival of OSCC patients. So, it is important to perform multivariate analysis, particularly for the combination of LLT1 expression in the stroma and tumor cells to verify whether LLT1 remains as an independent prognostic marker for OSCC.
2. The authors should correct the number of patients in the cohort. The correct number is 124 and not 125. Moreover, the authors should confirm the percentages on 2.1. It looks like they were based on 125 samples.
3. The readers of the study would benefit with a more detailed description of the cohort. The authors should describe the radiotherapy protocol (type of equipment and regimen) and the drugs used by the 14 patients submitted to chemotherapy. As the recruiting involved patients treated in 2000, therapeutic protocols have changed a lot. Another important aspect is related to recurrence. The authors should describe the type of recurrence and how it was determined. Were they histologically confirmed?
4. Many studies highlight the impact of histopathological features on prognosis of OSCC. In this sense, once available, the study would definitely have more impact if some of these features, including the status of surgical margins and presence of PNI and LVI, are explored.
5. Since immunohistochemical evaluation was performed by 2 observers, it is relevant to describe training, agreement level and situations of disagreement. By the end, the authors have grouped the samples in positive vs. negative, which is a strategy to describe, then not requiring the description of the features above.
6. A HR value without its 95% CI is useless. Moreover, for combination, both HR and 95% CI are missing.
7. In a context where most OSCC patients receive maximum treatment intensity possible, a parameter that predicts risk of recurrence is very likely to be useful clinically. As the number of recurrences reached ~43% of the cohort, the authors might consider checking for LLT1 impact on disease (recurrence)-free survival.
8. Table 1. Replace please G status for WHO histopathological grade.
Author Response
REVIEWER # 2
It was a pleasure to read this interesting and well-conducted study exploring the prognostic impact of LLT1 expression in OSCC. I have one major comment and other minor ones in order to improve the manuscript.
Response: We thank the reviewer for the positive comments about our work and for all the insightful suggestions.
- The lack of multivariate analysis is the major issue of this study. KM curves and univariate analysis are shallow for today’s standard. LLT1 was not associated with any of the clinicopathological parameters of the tumors, but the impact of them on survival was not assessed. On the other hand, LLT1 was significantly associated with the infiltration of some immune cells. Several studies have demonstrated that the composition of the inflammatory infiltrate is significantly associated with survival of OSCC patients. So, it is important to perform multivariate analysis, particularly for the combination of LLT1 expression in the stroma and tumor cells to verify whether LLT1 remains as an independent prognostic marker for OSCC.
Response: The following information has been added to the revised manuscript:
Patients with T3-T4 tumors, as well as patients with neck lymph node metastasis, and III and IV clinical stages exhibited a significantly worse disease-specific survival (p = 0.001, p = 0.01, and p = 0.002; respectively). Although patients with well-differentiated tumors showed a better survival, no significant differences in survival rates were observed among the different grades of tumor differentiation. Multivariable Cox regression analysis, including T classification (T1-T2 vs. T3-T4), N classification (N0 vs. N+), and LLT1 expression in TILs showed that the parameters independently associated with worse disease-specific survival were T3-T4 classification (HR = 2.55; 95% CI = 1.47–4.42; p = 0.001), and neck lymph node metastasis (HR = 1.89; 95%CI = 1.10–3.25; p = 0.02). However, LLT1 expression in TILs did not retain its significant association with a better prognosis (p = 0.09). Therefore, LLT1 expression was not found an independent prognostic marker.
- The authors should correct the number of patients in the cohort. The correct number is 124 and not 125. Moreover, the authors should confirm the percentages on 2.1. It looks like they were based on 125 samples.
Response: The number of patients as well as the percentages have now been corrected. We thank the reviewer for noting these errors.
- The readers of the study would benefit with a more detailed description of the cohort. The authors should describe the radiotherapy protocol (type of equipment and regimen) and the drugs used by the 14 patients submitted to chemotherapy. As the recruiting involved patients treated in 2000, therapeutic protocols have changed a lot. Another important aspect is related to recurrence. The authors should describe the type of recurrence and how it was determined. Were they histologically confirmed?
Response: Following the reviewer’s comment, further details about the studied patient cohort have been added in this new version of the manuscript (Results subsection 2.1.), as follows: From the total cohort of 124 patients, 75 were administered adjuvant radiotherapy with radiation doses in the range of 66–70 Gy, delivered in daily fractions of 1.9–2.1 Gy/fraction once daily. Fourteen patients with high risk features (positive or close margins or extranodal extension) received subsequent adjuvant chemotherapy with regimens that had two or three drug combinations, comprising a platinum-based agent (cisplatin or carboplatin) and either paclitaxel/docetaxel with or without 5-fluorouracil. The number of cycles ranged from one to six. Two patients also received cetuximab and one patient nivolumab.
Recurrences were located in the oral cavity or in the neck, and all of them were histologically confirmed.
- Many studies highlight the impact of histopathological features on prognosis of OSCC. In this sense, once available, the study would definitely have more impact if some of these features, including the status of surgical margins and presence of PNI and LVI, are explored.
Response: Here we evaluated the associations of LLT1 expression in both tumor cells and TILs with the clinicopathological variables and patients` outcome. Among them, positive surgical margins (in 11 cases) as well as the presence of PNI and LVI were quite small in number and in all of these cases radiotherapy and chemotherapy were added to the treatment. These variables failed to provide any relevant information in our studied OSCC cohort, and hence they were excluded from the analysis.
- Since immunohistochemical evaluation was performed by 2 observers, it is relevant to describe training, agreement level and situations of disagreement. By the end, the authors have grouped the samples in positive vs. negative, which is a strategy to describe, then not requiring the description of the features above.
Response: The expression levels were independently scored by two observers blinded to clinical information. A scoring system based on staining intensity was applied, and LLT1 expression was classified as negative staining (score 0), weak (score 1), moderate (score 2), or strong staining (score 3). There was a high inter-observer agreement (>95%). For statistical purposes, staining scores were dichotomized as negative expression (score 0) versus positive expression (scores 1 to 3). There were no disagreements between the two observers when the expressions were finally dichotomized.
- A HR value without its 95% CI is useless. Moreover, for combination, both HR and 95% CI are missing.
Response: We fully agree. 95% CI together with HRs have now been included.
- In a context where most OSCC patients receive maximum treatment intensity possible, a parameter that predicts risk of recurrence is very likely to be useful clinically. As the number of recurrences reached ~43% of the cohort, the authors might consider checking for LLT1 impact on disease (recurrence)-free survival.
Response: We completely agree. Following the reviewer’s comment, we have analyzed the impact of LLT1 on the risk of recurrence (i.e. disease-free survival) using Kaplan-Meier analysis. Patients harboring LLT1-positive tumors showed a lower mean disease-free survival (DFS) 92.98 (95% CI = 70.59 – 115.37) than those with LLT1-negative tumors (mean DFS 136.93; 95% CI = 113.35 – 160.51). However, differences were not statistically significant (Log-rank test, p = 0.26). On the other hand, patients harboring tumors with LLT1-positive TILs exhibited a slightly higher mean DFS 129.89 (95% CI = 103.18 – 156.60) compared to those with LLT1-negative TILs (mean DFS 124.28; 95% CI = 96.53 – 152.03; Log-rank test, p = 0.59). Therefore, unfortunately, LLT1 expression did not show a clinical utility as recurrence risk predictor in this OSCC cohort. Nevertheless, these data reinforce our findings regarding the prognostic relevance of tumoral and stromal LLT1 on OSCC patient survival, further supporting a relationship of tumoral LLT1 with poor survival rates and a possible immunosuppressive role, whereas stromal LLT1 expression in TILs is related to favorable outcomes in OSCC and an immune-active highly infiltrated TIME.
- Table 1. Replace please G status for WHO histopathological grade.
Response: G status has been replaced with WHO histopathological grade, as suggested.
Round 2
Reviewer 2 Report
Comments and Suggestions for Authors
The authors have adequately addressed my comments. I congratulate them for this excellent work.